# Parental Educational Attainment on Adolescent Educational Development: A Multi-Group Analysis of Chinese Left-Behind and Non-Left-Behind Children

**DOI:** 10.3390/bs14100870

**Published:** 2024-09-25

**Authors:** Guilin Xu, Chunhua Fu

**Affiliations:** 1School of Marxism, Wuhan Textile University, Wuhan 430200, China; glxu@wtu.edu.cn; 2School of Education, Minzu University of China, Beijing 100086, China

**Keywords:** left-behind children, intergenerational transmission, educational expectations, educational inequality

## Abstract

Utilizing data from the China Family Panel Studies (CFPS), this research focuses on the differences and mechanisms of intergenerational educational transmission between left-behind and non-left-behind children using structural equation modeling and multi-group path analysis. The findings indicate that significant intergenerational educational transmission effects exist in both groups, with parental educational attainment significantly impacting children’s academic performance. Further mediation analysis demonstrates that parental educational attainment influences children’s academic performance through the chain mediation effects of parental educational expectations and self-educational expectations. Multi-group path analysis reveals distinct mechanisms affecting academic performance: among non-left-behind children, parental educational attainment exerts a stronger direct influence on academic performance, while self-educational expectations are more influential for left-behind children. Additionally, the path coefficients for the effects of parental educational expectations and self-educational expectations on academic performance are higher for left-behind children than for non-left-behind children. Consequently, educational interventions should focus on enabling parents of left-behind children to effectively convey educational expectations through indirect methods (e.g., phone or online communication) and on enhancing left-behind children’s self-educational expectations through school and community support to facilitate academic achievement in the absence of direct family support.

## 1. Introduction

In recent years, the intergenerational transmission of education has attracted considerable public attention, often leading to discussions about the potential for children from disadvantaged backgrounds to achieve success [1]. Educational equity is intricately linked to household income growth, social mobility, and even broader economic development and social stability [2]. Education can act as a double-edged sword: while it can perpetuate social advantages for the privileged, it can also provide a pathway for ordinary families to achieve upward mobility. Whether education contributes to social stratification or promotes social mobility largely depends on the strength of its intergenerational transmission effects [3]. The Chinese government has placed substantial emphasis on educational equity, and the introduction of universal compulsory education has facilitated the widespread adoption of basic education. However, disparities in educational quality between urban and rural areas emerge early in basic education, leading to unequal opportunities and quality in high school and university admissions due to the cumulative effects of education [4]. Left-behind children, or liushou ertong in Chinese, first appeared in the academic literature in the early 1990s. In 2016, the Chinese government redefined LBC as children under 16 with both parents migrating to cities for work or one parent is absent, but the other is incapable of supervising the child [5]. These children, a particularly vulnerable group, face numerous challenges in their educational development. With rapid urbanization and large-scale migration from rural areas to cities in China, the number of migrant workers has increased substantially. Due to the constraints of the household registration system and economic conditions, many migrant workers are compelled to leave their children behind in rural areas, resulting in a significant population of left-behind children. According to statistics, there were 66.93 million left-behind children aged 0–17 in China in 2020 [6]. The absence of one or both parents in their upbringing has heightened concerns across society regarding the issues faced by left-behind children and the challenges to their development. Considering the profound impact of the intergenerational transmission of education, it is crucial to examine whether differences exist in this transmission between left-behind and non-left-behind children in China.

Intergenerational transmission of education pertains to the impact of parents’ educational levels on their children’s educational outcomes. This transmission encompasses direct educational support, the provision of educational resources, and the conveyance of educational expectations [7,8]. Parental educational expectations play a crucial mediating role in how family background influences students’ academic performance and educational achievements [9,10]. Research indicates that in economically disadvantaged families, parents’ expectations and their engagement with their children’s education are pivotal factors affecting whether children persist in their studies or drop out [11]. Moreover, high educational expectations from parents in low-income families significantly contribute to their children’s academic success and social mobility [12]. Hence, the question arises: do educational expectations serve as a mediator in the intergenerational transmission of education?

This study aims to investigate the impact of parental educational inequality on the educational development of children from left-behind and non-left-behind families, with a particular focus on the roles played by parental and self-educational expectations in the intergenerational transmission of education. Previous research has predominantly concentrated on urban-rural differences, indicating that intergenerational transmission of education is more pronounced in rural areas compared to urban areas, with a widening disparity [13,14]. However, there is a notable lack of studies specifically addressing the intergenerational transmission of education among left-behind children and even fewer comparative studies between left-behind and non-left-behind children. This study builds upon existing research by utilizing data from the 2020 China Family Panel Studies to analyze how parental education levels influence children’s academic performance and the underlying mechanisms. It seeks to determine whether there are differences in the intergenerational transmission of education between left-behind and non-left-behind families and to understand the role of educational expectations in this process. The findings of this study will have significant policy implications for understanding the mechanisms of intergenerational transmission, addressing educational equity for left-behind children, and overcoming class-based barriers to enhance social mobility.

## 2. Literature Review

### 2.1. Parental Educational Attainment and Adolescent Educational Development

Parental educational attainment refers to the highest level of education completed by a parent. The theory of Intergenerational Transmission of Education suggests that parents’ educational attainment profoundly influences their children’s educational development. A wealth of empirical research supports this theory, demonstrating a notable intergenerational transmission effect in the educational achievements of both parents and their offspring [15,16]. This concept fundamentally entails parents utilizing various forms of capital, such as economic, social, and cultural capital, to aid their children in achieving higher educational qualifications, which are deemed institutionalized cultural capital. This process is mediated through two principal mechanisms. The first is the resource conversion mechanism, where families with higher parental education levels leverage their economic and social advantages to offer superior educational resources to their children. Kaushal et al. (2011) observed that parents with elevated educational levels tend to have higher developmental aspirations, a greater willingness to engage in education, and a higher inclination to invest in educational endeavors, thereby boosting their motivation for educational investment [17]. The second mechanism is cultural reproduction. Parents with higher levels of education are more capable of supporting and improving their children’s academic performance, which in turn provides their children with greater access to better educational opportunities [18]. Furthermore, these parents, having reaped the benefits of education themselves, unintentionally transmit their educational values to the next generation through their behaviors, attitudes, and everyday interactions, thereby facilitating the intergenerational transmission of education [19].

Consequently, this study hypothesizes a significant positive relationship between parental educational attainment and adolescent academic performance.

### 2.2. The Mediating Role of Parental Educational Expectations and Self-Educational Expectations 

Research has demonstrated that parental educational expectations serve as a partial mediator in the relationship between parental educational attainment and adolescent educational development. Elevated parental education levels typically correlate with higher educational expectations for their children, which, in turn, foster the children’s academic achievements [8]. Furthermore, the educational expectations of significant others, such as parents and peers, can influence how family background characteristics affect an individual’s attainment of social status [20]. Social mobility is not only achieved through socio-psychological mechanisms but is also facilitated by these mechanisms. For example, parents who maintain optimistic educational expectations are likely to assist their children in surpassing academic success levels predicted solely by the family’s socioeconomic status. Conversely, parents with pessimistic educational expectations may observe lower levels of achievement among their children [21]. From this perspective, children’s educational trajectories and their engagement with the educational system reflect the impact of their parents’ underlying beliefs. As elucidated by Malczyk and Lawson (2019), parental expectations rank among the most influential parenting practices, surpassing the effects of parent-child communication, family structure, and parental involvement in schools [22]. Consequently, this study posits that parental educational expectations mediate the relationship between parental educational attainment and adolescent academic performance in both left-behind and non-left-behind children.

Parental education attainment and expectations are significant external factors influencing adolescent academic development, while self-educational expectations highlight the importance of internal motivation. According to social cognitive theory, adolescents’ beliefs about their abilities and expectations for their own academic achievement play a crucial role in shaping their behaviors and performance [23]. Central to this theory is the concept of self-efficacy—the belief in one’s ability to succeed—which is a critical driver of academic behaviors and outcomes. Students with higher self-educational expectations tend to set ambitious goals, persist through challenges, and actively seek support when needed [24]. Thus, self-educational expectations act as an essential mediator in the link between parental educational attainment and academic success. Research indicates that parental educational levels significantly influence their children’s self-educational expectations. Parents with higher educational levels are generally more effective in fostering their children’s educational aspirations and providing the necessary learning resources and support [7]. Adolescents’ educational expectations are critical predictors of their future academic success, including test scores, school performance, and dropout rates [25]. Studies have shown that, regardless of urban or rural settings or different school types, children’s educational expectations significantly impact their educational attainment. Thus, educational expectations are regarded as robust predictors of academic achievement [21]. Consequently, this study hypothesizes that self-educational expectations significantly positively predict adolescent academic performance and mediate the relationship between parental educational attainment and academic success.

Building on the theoretical foundation, parental educational attainment can impact adolescents’ academic performance through both parental educational expectations and self-educational expectations across left-behind and non-left-behind families. While parental educational expectations are positively associated with adolescents’ academic achievements, they might not exert a direct influence [26]. Research suggests that adolescents’ academic success is mediated by their own educational expectations, indicating that self-educational expectations serve as a mediator [27]. The effect of parental expectations on adolescents’ engagement in learning represents a potential transmission process. Rutchick et al. (2009) found that parents’ educational expectations when children are 6 years old are closely linked to the children’s self-educational expectations at age 13, with self-educational expectations mediating the relationship between parents’ expectations and children’s academic outcomes [28]. Thus, this study hypothesizes that in both intact families and among left-behind children, parental educational expectations and self-educational expectations play a chain mediation role between parental educational attainment and adolescent academic performance.

### 2.3. The Moderating Role of Left behind Experience

It is important to examine whether there are differences in adolescent academic performance, as well as in parental educational levels, educational expectations, and self-educational expectations, between left-behind children and non-left-behind children. The Wisconsin Theory Model has significantly contributed to understanding the intermediary mechanisms through which family background affects educational attainment, emphasizing that parental educational expectations are a crucial mediating variable in this model. Research indicates that higher family socioeconomic status is associated with higher parental expectations for their children, which in turn leads to greater academic achievements and increased chances of higher education. This finding has been empirically validated in various countries [9,29,30]. Conversely, does a lower socioeconomic status also result in lower parental and self-educational expectations? For left-behind children, parental educational expectations under the circumstances of parental migration have unique characteristics. On one hand, left-behind children, due to physical and temporal separation, often receive less direct educational support from their parents. On the other hand, many parents, dissatisfied with their own educational levels, income, and social status, strive to provide better economic conditions and educational opportunities for their children through migration and thus often have higher educational expectations for them [31]. Additionally, the rise of online media has opened new avenues for migrant parents to engage in their children’s education and communicate their expectations. Mobile communication tools like video calls, messaging apps, and online platforms help bridge the emotional and physical gap between parents and left-behind children, transcending time and space to maintain family bonds. This sense of ‘virtual co-presence’ [32] fosters emotional support and solidarity [33] and eases the emotional toll of separation [34], enabling children to feel connected despite the distance [35]. Moreover, these digital platforms allow parents to actively engage in their children’s education, expressing expectations and providing support. Research indicates that this involvement enhances both children’s academic performance and emotional well-being [36]. Therefore, this study hypothesizes that there are no significant differences in parental educational expectations and self-educational expectations between left-behind and non-left-behind children.

Moreover, there may be differences in academic performance between adolescents from different groups and how this performance relates to parental educational attainment and educational expectations. Compared to non-left-behind children, parents of left-behind children typically have lower educational levels. However, according to the self-fulfilling prophecy theory, expectations can have a stronger self-fulfilling effect in relatively disadvantaged groups [37]. Specifically, for rural left-behind children, migrant parents can offset the negative impacts of lower economic conditions and physical separation by providing more encouragement and high expectations, which can enhance their resilience and development [38]. Therefore, the impact of parental educational attainment and educational expectations on the academic development of left-behind children versus those from non-left-behind families may differ. The academic development of left-behind children is likely to be more influenced by parental educational expectations, functioning as a motivational factor, while for non-left-behind children, their academic outcomes may be more directly associated with the intergenerational transmission of parental educational attainment. This reflects the differential mediation effects of parental expectations and self-expectations on the relationship between parental educational attainment and academic performance, as outlined in the hypotheses.

In summary, there may be differences between left-behind and non-left-behind children in terms of parental educational attainment, academic performance, parental educational expectations, and self-educational expectations, as well as in the interrelations of these factors. This study proposes the following hypotheses:

**Hypothesis** **1:**
*There are differences between left-behind and non-left-behind children regarding parental educational attainment, academic performance, parental educational expectations, and self-educational expectations.*


**H1.1:** 
*The parental educational attainment and academic performance of left-behind children are significantly lower than those of non-left-behind children.*


**H1.2:** 
*There are no significant differences in parental educational expectations and self-educational expectations between left-behind and non-left-behind children.*


**Hypothesis** **2:**
*Both left-behind and non-left-behind children’s parental educational expectations and self-educational expectations mediate the relationship between parental educational attainment and academic performance through a chain mediation effect.*


**Hypothesis** **3:**
*The chain mediation effects of parental educational expectations and self-educational expectations on the relationship between parental educational attainment and academic performance differ between left-behind and non-left-behind children.*


## 3. Method

### 3.1. Data Source and Participants

This study utilizes data from the 2020 China Family Panel Studies (CFPS) conducted by the China Social Science Survey Center at Peking University, integrating key explanatory variables from 2016 and 2018. The CFPS includes extensive surveys covering communities, families, adults, and children, sampling randomly from 25 provinces, municipalities, and autonomous regions. The dataset effectively represents 95% of China’s demographic, reflecting multifaceted conditions across social, economic, demographic, educational, and health dimensions, thus offering profound national representativeness and scientific research utility. For the analysis sample, the study followed these steps: First, adolescents aged 10–15 were selected as the study subjects because their self-reported questionnaires included relevant indicators such as academic performance and parental educational involvement, and they have stronger screening criteria compared to younger primary school students. Second, by matching child IDs with parent questionnaires and excluding samples with incomplete data, the final sample size for the model analysis was 1039. 

Among the sample, there were 117 left-behind children, accounting for 11.3%, while 922 were non-left-behind children, making up 88.7% of the sample. In terms of gender, 480 participants were male (46.2%) and 559 were female (53.8%). The majority of participants held rural household registrations (847 individuals, 81.5%), while 192 (18.5%) had urban household registrations. Regarding family size, 126 participants (12.1%) were from one-child families, 628 (60.4%) were from families with two children, and 285 (27.4%) were from families with three or more children.

### 3.2. Operational Definition of Variables

The dependent variable in this study is the academic achievement of adolescents, measured by the average level of their Chinese and mathematics grades as reported by their parents for the previous semester in 2020. Information on other subjects is not available in the CFPS. Parent-reported academic performance has been used in the previous literature to examine the proposed mediation hypothesis (e.g., Spera, Wentzel, & Matto, 2009; Zhan & Sherraden, 2003) [39,40]. For ease of interpretation, we treat these two variables as continuous in analyses. The grading scale is as follows: 1 signifies poor, 2 signifies average, 3 signifies good, and 4 signifies excellent. 

Independent Variables. Building on the methodologies employed by Steinmayr et al. (2010) [41] and Tang J (2016) [42], parental educational attainment is measured by the highest level of education achieved by the parents. The conversion from years of education to educational levels is defined as follows: 0 = illiterate, 6 = elementary school, 9 = junior high school, 12 = high school, 15 = associate degree, 16 = bachelor’s degree, 19 = master’s degree, and 23 = doctoral degree.

Mediating Variables. Parental educational expectations reflect the level of education that parents anticipate for their children. This variable is assessed through 8 options, with the levels of education converted as follows: 0 = no formal education, 6 = elementary school, 9 = junior high school, 12 = high school, 15 = associate degree, 16 = bachelor’s degree, 19 = master’s degree, and 23 = doctoral degree. A similar single-item measure of parental educational expectations has been widely used in studies with youth (e.g., Froiland et al., 2013a; Xu et al., 2010; Zhang et al., 2011) [9,43]. This measure has demonstrated strong reliability and predictive validity, with parent expectations in early childhood predicting both later expectations and academic achievement across various subjects in longitudinal studies (Froiland et al., 2013a) [9]. Self-educational expectations refer to the level of education that students aspire to achieve. The scale follows the same categorical structure as parental educational expectations, with higher values corresponding to higher educational aspirations.

Moderator Variables. To distinguish between left-behind families and single-parent families, this study categorizes children based on two indicators: parental marital status and the living arrangements of the respondents with their parents. Accordingly, children are divided into two groups: non-left-behind children (both parents are married, and neither parent is away) and left-behind children (both parents are married, with one or both parents being away).

The control variables include four factors: gender (male = 1), age (ranging from 10 to 15 years), household registration (agricultural household registration = 1), and number of children (1 = only child, 2 = two children, 3 = three or more children).

### 3.3. Data Analysis

The analysis of the data proceeded in three steps. Initially, SPSS 25.0 was used to perform descriptive statistics and correlation analysis, providing insights into the overall composition of adolescent groups and the interrelations between adolescent groups, parental educational attainment, parental educational expectations, and self-educational expectations. Subsequently, Amos 24.0 was utilized to construct a structural equation model and test the mediation model, aiming to assess the chain mediation effect of parental educational expectations and self-educational expectations in the relationship between parental educational attainment and academic performance. Finally, a multi-group analysis was conducted to explore the differences in mediation model paths among different adolescent groups.

## 4. Results

### 4.1. Common Method Bias Test

To assess potential common method bias, this study employed Harman’s single-factor test. In this test, all items are subjected to an unrotated factor analysis, and the results help to determine whether a single factor accounts for the majority of the variance in the data. The results revealed that only one factor had an eigenvalue greater than 1, and the first factor accounted for 44.63% of the variance, which is below the 50% threshold. Therefore, the study does not exhibit significant common method bias, suggesting that the relationships observed in the study are less likely to be artificially inflated due to measurement method bias [44]. 

### 4.2. Differences between Left-Behind and Non-Left-Behind Children in Terms of Parental Educational Attainment, Parental Educational Expectations, Self-Educational Expectations, and Academic Performance

Table 1 presents an analysis of the differences between left-behind and non-left-behind children in various variables. The findings reveal that: First, the educational attainment of parents of left-behind children is significantly lower than that of parents of non-left-behind children (t = 5.32, *p* < 0.001). Second, there are no significant differences between left-behind and non-left-behind children regarding parental educational expectations, self-educational expectations, or academic performance. This suggests that despite their parents’ lower educational levels, left-behind children’s educational expectations and academic achievements do not lag behind those of their non-left-behind peers. This result challenges the conventional belief that left-behind children perform worse academically, highlighting the need for further research into the factors that influence their academic development. Hypothesis 1 is supported.

Additionally, significant differences were observed between left-behind and non-left-behind families in terms of the number of children, household registration, and family income. Left-behind families tend to have more children on average (t = 0.39, *p* < 0.01), reflecting cultural and economic differences in rural areas. A significantly higher proportion of left-behind children have rural household registration compared to their non-left-behind peers (t = −2.98, *p* < 0.01), consistent with the higher prevalence of economic migration in rural populations. Family income also differs significantly (t = 0.51, *p* < 0.01), with left-behind families experiencing lower and more unstable income, likely due to the income instability of migrant worker families.

### 4.3. Correlation Analysis of Parental Educational Attainment, Parental Educational Expectations, Self-Educational Expectations, and Academic Performance

A correlation analysis was performed on adolescents’ parental educational attainment, parental educational expectations, self-educational expectations, academic performance, and other demographic variables. According to Table 2, there are significant correlations among these four variables: parental educational attainment, parental educational expectations, self-educational expectations, and academic performance. These significant correlations meet one of the key conditions for testing mediation effects. Moreover, while the variables are significantly related, the correlation coefficients do not exceed 0.8, suggesting that multicollinearity is not a major concern in this analysis.

### 4.4. Chain Mediation Effects of Parental Educational Expectations and Self-Educational Expectations in the Relationship between Parental Educational Attainment and Academic Performance

To address potential multicollinearity issues, all variables were standardized, and a path analysis model was developed. The results indicated that the model was saturated. According to Kline (2023), even in a saturated model, important relationships between variables can be identified through the analysis of path coefficients and effect sizes [45]. The specific path coefficients for the model are illustrated in Figure 1.

The study employed the bias-corrected percentile bootstrap method to test for mediation effects with 5000 resamples. If the confidence interval (CI) for the indirect effect does not include zero, the indirect effect is considered statistically significant, indicating that a mediation effect is present [46]. The results are shown in Figure 1. Parental educational attainment has a significant positive impact on adolescents’ academic performance (β = 0.17, *p* < 0.001). The multiple mediation effects between parental educational attainment and academic performance are detailed in Table 3. Specifically, the mediation effect of parental educational attainment through parental educational expectations on academic performance (β = 0.032, 95% CI: [0.018–0.051]) accounts for 13.8% of the total effect. The mediation effect of parental educational attainment through self-educational expectations on academic performance (β = 0.025, 95% CI: [0.014–0.041]) constitutes 2.5% of the total effect. The chain mediation effect of parental educational attainment through both parental educational expectations and self-educational expectations on academic performance (β = 0.009, 95% CI: [0.005–0.016]) makes up 3.88% of the total effect. Based on these findings, Hypothesis 2 is supported.

### 4.5. Investigating Differences in the Chain Mediation Model Paths between Left-Behind and Non-Left-Behind Children: A Multi-Group Analysis

To investigate the differences in the chain mediation model between left-behind and non-left-behind children, this study conducted a multi-group path analysis using left-behind experience as the grouping variable. The analysis tested model fit from the least restrictive to the most restrictive models. The results showed that the fit of the unconstrained model M1 (unconstrained), also known as the baseline model (baseline model), structural weights model M2 (structural weights), structural covariances model M3 (structural covariances), and structural residuals model M4 (structural residuals) was generally acceptable. Since this study mainly focuses on the moderating effect of family type on the impact mechanisms of parental educational attainment, parental educational expectations, and self-educational expectations on academic performance in adolescents, the analysis primarily examined the invariance of the structural weights model between the two family structures. The chi-square difference test was used to compare the unconstrained model with other constrained models. The nested model comparison results indicated that assuming the baseline model is correct, there was no significant difference between the model with equal structural coefficients and the most restrictive model (ΔX^2^ = 4.51, ΔDF = 6, *p* > 0.05). This means that the structural weights model maintains invariance between left-behind and non-left-behind children, indicating no significant differences in model coefficients. However, a detailed comparison of the path coefficients (see Table 4) revealed some differences in the relationships between parental educational attainment, educational expectations, and academic performance across left-behind and non-left-behind children. In non-left-behind children, parental educational attainment has a significant positive impact on parental educational expectations, self-educational expectations, and academic performance. In contrast, parental educational attainment does not significantly impact these variables in left-behind children. Nevertheless, among left-behind and non-left-behind families, parental educational expectations and self-educational expectations significantly positively affect academic performance. Thus, Hypothesis 3 is partially supported.

## 5. Discussion

### 5.1. Differences in Parental Educational Attainment, Parental Educational Expectations, Self-Educational Expectations, and Academic Performance

The findings indicate that parents of left-behind children have significantly lower educational attainment compared to parents of non-left-behind children. This difference likely reflects the disparity in educational resources between rural and urban areas, as well as the generally lower educational levels of migrant workers. Despite slightly lower educational expectations for left-behind children, the difference is not statistically significant. This suggests that the common aspiration for children to succeed and excel through education is shared across both left-behind and non-left-behind families. Among left-behind and non-left-behind families, parents wish for their children to change their futures and achieve better development through education. In terms of self-educational expectations, those non-left-behind children have slightly higher expectations than those of left-behind children, but the difference is again not significant. This finding is consistent with previous studies [47], which demonstrated that despite the challenges posed by parental migration, many left-behind children remain diligent, self-reliant, and maintain high educational expectations. Regarding academic performance, there is no significant difference between the two groups, suggesting that, within the current sample, being a left-behind child does not directly impact academic performance significantly. Support from schools and other family members may help mitigate the barriers often experienced by parents who have a history of migration. This aligns with the Ecological Systems Theory, which proposes that individual development results from interactions within multi-layered environmental systems [48].

### 5.2. The Relationship between Parental Educational Attainment, Parental Educational Expectations, Self-Educational Expectations, and Adolescent Academic Performance

The empirical analysis of this study reveals that parental educational expectations and self-educational expectations serve as significant chain mediators between parental educational attainment and adolescent academic performance. Specifically, parental educational attainment impacts students’ academic performance directly and indirectly through the mediation effects of parental educational expectations and self-educational expectations. Higher parental education levels are associated with elevated parental educational expectations. According to cultural capital theory [49], parents with higher education possess greater knowledge, skills, and educational resources, enabling them to appreciate the value of education and set higher expectations for their children’s academic achievements. These expectations are conveyed to children through routine interactions and behaviors, thereby motivating them to excel academically.

Secondly, parental educational attainment also exerts an indirect effect on children’s academic performance through the mediating role of self-educational expectations. According to social cognitive theory, people’s behaviors and cognitive processes are shaped by observing the actions and outcomes of others, particularly significant figures like parents. Parents with higher educational levels serve as exemplary role models through their own educational accomplishments, thereby encouraging their children to set higher educational aspirations. Additionally, these parents are more likely to provide substantial academic support and encouragement, which enhances their children’s confidence and self-expectations regarding academic success. The mediating role of self-educational expectations illustrates that the influence of parental education on children’s development is not direct and straightforward but involves a motivational process. This influence operates through children’s cognitive processes, shaping and refining their self-educational expectations, which subsequently impacts their academic performance.

Moreover, parental educational attainment exerts a cumulative influence on children’s academic performance through the chain mediation effect of both parental and self-educational expectations. Consistent with previous studies, this finding demonstrates that while parental and self-expectations independently impact academic performance, their combined mediation effect is even more substantial [25]. More importantly, our analysis further reveals that the chain mediation effect accounts for 3.88% of the total mediation effect, surpassing the contribution of self-expectations when considered in isolation. This highlights the pivotal role of parental expectations, which not only directly influence children’s academic performance but also strengthen their academic outcomes by enhancing their self-motivation. These findings provide critical insights for the formulation of educational policy. By targeting both parental and self-educational expectations, it is possible to address academic inequality caused by intergenerational transmission. Specifically, for left-behind children, encouraging parents to establish higher educational expectations and promoting a focus on education can foster stronger self-educational aspirations, thereby narrowing the achievement gap between these children and their urban or more advantaged peers.

### 5.3. Differences in the Relationships between Parental Educational Attainment, Parental Educational Expectations, Self-Educational Expectations, and Academic Performance in Non-Left-Behind Children and Left-Behind Children

The group analysis results indicate no significant differences in the coefficients between left-behind children and non-left-behind children, suggesting that the mechanism through which parental educational attainment affects academic performance via parental educational expectations and self-educational expectations is consistent across both groups. However, a more detailed examination of the path coefficients reveals notable differences in the specific impact pathways.

For non-left-behind children, parental educational attainment has a stronger direct effect on children’s academic performance. In contrast, for left-behind children, self-educational expectations have a more pronounced effect on academic performance. This finding implies that left-behind children rely more on their internal motivation and personal educational goals to achieve academic success.

Furthermore, the path coefficients for the impact of parental educational expectations and self-educational expectations on academic performance are higher for left-behind children than for non-left-behind children. This aligns with the self-fulfilling prophecy effect, where the impact of expectations is more potent in disadvantaged groups [37]. Research also indicates that high educational expectations from parents are crucial for disadvantaged students to achieve upward social mobility [50]. This finding offers a fresh perspective on the study of educational intergenerational transmission. While there is some level of intergenerational transmission of educational attainment, the presence of high educational expectations from parents can mitigate the negative impacts associated with lower parental education levels and support children’s academic success. This insight is valuable for policy development, emphasizing the need to raise parental educational expectations for left-behind children, especially in the absence of direct family support.

It is important to note that, irrespective of family structure, parental educational expectations have a significant impact on children’s self-educational expectations. Therefore, educational interventions should focus on helping the parents of left-behind children convey their educational expectations through indirect methods, such as phone or online communication. Additionally, enhancing self-educational expectations through school and community support is crucial for left-behind children, helping them achieve academic success despite lacking direct family support.

## 6. Conclusions

Previous studies have investigated various mediating mechanisms by which parental educational attainment influences adolescent educational development, including parental educational expectations, educational involvement, and parent-child communication. These studies demonstrate that parental education plays a significant role in shaping adolescent outcomes, with educational expectations being a key factor in the intergenerational transmission of education. Additionally, research focusing on the population of left-behind children has highlighted the adverse effects of parental migration on their cognitive and psychological development. However, comparative analyses of the mediating effects in the educational transmission between left-behind and non-left-behind children remain limited.

This study conducts a cross-group comparison to explore the chain mediation effect of parental educational expectations and self-educational expectations in the relationship between parental educational attainment and adolescent educational outcomes. The findings underscore that, particularly for left-behind children, the effective transmission of parental educational expectations and the enhancement of self-educational expectations are critical in mitigating the negative impacts of educational transmission, thereby supporting the realization of their developmental potential. In contrast to previous research that predominantly adopts a single-group perspective, this study provides a more comprehensive empirical analysis, offering significant insights for the development of more targeted and group-specific educational policies. Moreover, it offers critical empirical evidence to support the educational advancement of left-behind children.

## 7. Limitations and Suggestions for Future Studies

First, this study uses a cross-sectional design, which clarifies the relationships between parental educational attainment, parental educational expectations, self-educational expectations, and academic performance but does not allow for conclusions about causality between these variables. The static nature of cross-sectional designs offers limited support for validating causal pathways within the chain mediation model. To more accurately uncover causal relationships, future research should adopt longitudinal designs that track students across various developmental stages. This approach will help to explore the dynamic changes and causal pathways among parental educational attainment, educational expectations, and academic performance, thereby providing stronger empirical support for the chain mediation model.

Moreover, this study primarily focused on the relationships between parental educational attainment, parental educational expectations, self-educational expectations, and academic performance. While the research indicates that educational expectations are a key factor in educational transmission, the educational development of left-behind children also relies on other forms of social support and compensatory mechanisms, such as the care provided by teachers and peer support. These factors emphasize the crucial role of social support systems in achieving educational equity. Future research should integrate these factors and explore their roles and effects in the educational intergenerational transmission process. This will further illuminate the complex mechanisms behind educational transmission and offer more scientifically based and effective policy recommendations and practical guidance for promoting educational equity and improving student academic performance.

## Figures and Tables

**Figure 1 behavsci-14-00870-f001:**
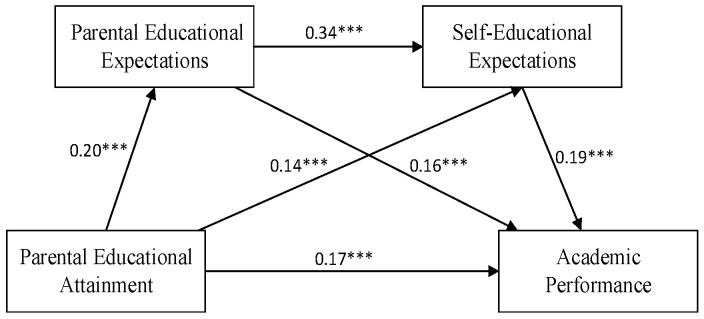
Path Coefficients of the Model. *** *p* < 0.001.

**Table 1 behavsci-14-00870-t001:** Differences in Parental Educational Attainment, Parental Educational Expectations, Self-Educational Expectations, and Academic Performance Between Left-Behind and Non-Left-Behind Families (x ± s).

Variable	Non-Left-Behind Children	Left-Behind Children	t
Parental Educational Attainment	9.82 ± 0.12	7.96 ± 0.31	5.32 ***
Parental Educational Expectations	15.93 ± 0.08	15.65 ± 0.24	1.17
Self-Educational Expectations	15.32 ± 0.10	14.91 ± 0.28	1.38
Academic Performance	2.74 ± 0.03	2.58 ± 0.09	0.55
Age	12.14 ± 1.79	11.99 ± 1.77	0.86
Gender	0.54 ± 0.50	0.52 ± 0.50	0.38
Number of Children	2.13 ± 0.61	2.30 ± 0.56	0.39 **
Household Registration	0.8 ± 0.40	0.9 ± 0.31	−2.98 **
Family Income	3.92 ± 1.62	3.49 ± 1.58	0.51 **

Note: ** *p* < 0.01, *** *p* < 0.001.

**Table 2 behavsci-14-00870-t002:** Correlations (r) Among Parental Educational Attainment, Parental Educational Expectations, Self-Educational Expectations, Academic Performance, and Demographic Variables.

	x ± s	1	2	3	4	5	6	7	8
1 Parental Educational Attainment	9.61 ± 3.62	1							
2 Parental Educational Expectations	15.90 ± 2.50	0.17 **	1						
3 Self-Educational Expectations	15.27 ± 2.98	0.22 **	0.37 **	1					
4 Academic Performance	2.73 ± 0.88	0.23 **	0.27 **	0.26 **	1				
5 Age	12.13 ± 1.79	−0.13 **	−0.02	−0.11 **	−0.16 **	1			
6 Gender	0.54 ± 0.50	−0.03	0.01	−0.03	−0.09 **	0.01	1		
7 Number of Children	2.15 ± 0.61	−0.32 **	−0.13 **	−0.15 **	−0.09 **	0.00	−0.14 **	1	
8 Household Registration	0.82 ± 0.39	−0.40 **	−0.10 **	−0.14 **	−0.08 *	0.04	−0.00	0.23 **	1
9 Family Income	3.87 ± 1.62	0.41 **	0.10 **	0.13 **	0.11 **	−0.07 *	−0.01	−0.15 **	−0.27 **

Note: * *p* < 0.05, ** *p* < 0.01.

**Table 3 behavsci-14-00870-t003:** Mediation Effects of Parental Educational Expectations and Self-Educational Expectations between Parental Educational Attainment and Academic Performance.

Path	Std. Path Coeff.	Effect Size/%	95% CI	
			Lower	Upper
Parental Educational Attainment → Parental Educational Expectations → Academic Performance	0.032	13.79	0.018	0.051
Parental Educational Attainment → Self-Educational Expectations → Academic Performance	0.025	10.78	0.014	0.041
Parental Educational Attainment → Parental Educational Expectations → Self-Educational Expectations → Academic Performance	0.009	3.88	0.005	0.016
Total Mediation Effect	0.066	28.45	0.011	0.022
Direct Effect	0.166	71.55	0.027	0.054
Total Effect	0.232	100.0	0.043	0.079

**Table 4 behavsci-14-00870-t004:** Comparison of Coefficients in the Chain Mediation Model.

Path	β	
	Non-Left-Behind Children	Left-Behind Children
Parental Educational Attainment → Parental Educational Expectations	0.194 ***	0.148
Parental Educational Attainment → Self-Educational Expectations	0.137 ***	0.166
Parental Educational Attainment → Academic Performance	0.192 ***	0.013
Parental Educational Expectations → Self-Educational Expectations	0.345 ***	0.338 ***
Parental Educational Expectations → Academic Performance	0.163 ***	0.271 **
Self-Educational Expectations → Academic Performance	0.153 ***	0.220 *

Note: * *p* < 0.05, ** *p* < 0.01, *** *p* < 0.001.

## Data Availability

The data that support the findings of this study are available from the corresponding author upon reasonable request.

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
