# Peer review of "Parental Educational Attainment on Adolescent Educational Development: A Multi-Group Analysis of Chinese Left-Behind and Non-Left-Behind Children"

_behavsci, 2024, doi:10.3390/bs14100870_

Round 1

Reviewer 1 Report

Comments and Suggestions for Authors

The topic is interesting theoretically and for the practical implications of the results in educational terms. However, the authors’ attention is oriented more toward providing information on the quantitative analyses while basic theoretical concepts and other details are briefly mentioned or absent. 

Page 2, line 85,   the concept of attainment is not defined.

Page 1, line 43. Please define "left-behind children"

Page 3, line 100. Please clarify. Do you mean are capable of? Or?

Page 3, line 102. Do you mean intentionally or not? Please clarify and support your position or refer to the explanations provided in the literature

Page 3, line 102: Subtly: what is the meaning the authors intend to convey with this adjective?

Page 3, 130 Please explain more in detail the reason for setting  self-educational expectations as mediators between parental educational attainment and academic success

Page 4, lines 174-176. Given the relevance of the paper, please motivate with study results and add references to support this statement.

Page 4, lines 190-192. Please, align the language used in developing the statements so that they are more closely related to and  align with the hypotheses  

Page 5, line 214 _ This paragraph should include a detailed description of the participants and their personal and contextual characteristics

Additionally, in a separate paragraph, the authors should describe the tools used, the dimensions addressed, and the basic psychometric characteristics of the questionnaires, together with details on the authors of the tools and  eventually on their previous use.

Page 6, line  258. The heading refers to Measures although it refers to the analyses. The heading of paragraphs introducing the choices regarding the analyses usually are different

Page 6. Line 269. Common method bias test: please give more space to explaining the meaning of the results of this test

Page 9, line 335. Please, as for the other paragraphs, besides naming the analysis conducted,  highlight in the heading the aim of the analysis.

Additionally, since the paper refers to left behind experience, I suggest the authors stay at this level throughout the paper insofar as what distinguishes the two “groups” in this analysis is again the same variable ( that is left behind experience) and thus avoiding moving to a different level of analysis  or using left behind experience to identify family “types”.  

Page 10, line 374. Here, as  sometimes throughout the study, authors  mention families   

Page 10, line 384. I suggest mentioning not the negative effects of parental migration but referring to barriers often experienced by parents who have a history of migration

Page 10, from line 387. Are these hypotheses developed by the authors? Please make it explicit.  Is there any data that the authors can refer to here to support their position?

Page 10, line 402. Please refer to the authors and published papers here

Page 10, line 420. The authors provide their interpretation of the results here. The readers may benefit from a more direct link to the results obtained in this study and studies available in the literature.

Conclusions. Here, the authors summarize previous findings and provide a summary of the key findings. Please visually distinguish ( new line?) where this second part starts. 

Comments on the Quality of English Language

The paper is quite easy to follow,  only a general linguistic check is suggested (see review comments)

Reviewer 2 Report

Comments and Suggestions for Authors

I think this is a really interesting use of a large secondary dataset to answer a very current and pressing issue. However, there are a few areas of interest I would like to see addressed. 

1. Although the literature review has clear sub-headings there is a lot of overlap within the sections. I was wondering if this could be refined or clarified (with signposting within each section) a little more?

2. I would like to know a little more about the CFPS, how and what data is collected.

3. Associated with this I think there could be more detail on the sample in terms of perhaps a table presenting the overall profile, including number of left-behind and number not left-behind.

4. I may be mistaken but my understanding is that left-behind children tend to be in rural areas so I wonder why the analysis does not focus on left-behind and not left behind in this (i.e. rural) context. I think some explanation/justification would be useful, especially as Household registration, which I think relates to this status is significantly correlated with the other variables in Table 2.

5. line 248 I am curious about whether or not educational expectations is actually a continuous variable given that there are 8 options (and they are not evenly spaced so not interval or ratio) i.e. there are specified points on the scale?

6. Section 3.3 measures - should be in the past tense as the analysis has been conducted?

7.  Discussion: ll428-431 ' Consequently, it is imperative to encourage parents of left-behind children to place greater emphasis on their children's education, elevate their educational expectations, and foster higher educational aspirations in their children.' Linked to this perhaps in future research there could be something more about factors that help or hinder parents to do this - I suspect it is not because parents do not wish to but there are actually structural (rather than individual) factors that make this more difficult? 

Reviewer 3 Report

Comments and Suggestions for Authors

Dear authors,

Thank you for trusting me to review your paper.

I found i very interesting, and my decision is "Accept after minor revisions", as I detected only two issues that could be solved to strengthen the view of your paper.

Firstly, in Table 1 you can keep only the significant results, as you already present the full results in the following paragraph. This can make your Table more understandable with only significant items.

Secondly, please indicate the statistically significant paths in Table 3. You have a footnote, but there are no pointed clearly.

Congratulations, again! Keep up the good work!
